# The Role of Epigenetics in Primary Biliary Cholangitis

**DOI:** 10.3390/ijms23094873

**Published:** 2022-04-28

**Authors:** Alessio Gerussi, Elvezia Maria Paraboschi, Claudio Cappadona, Chiara Caime, Eleonora Binatti, Laura Cristoferi, Rosanna Asselta, Pietro Invernizzi

**Affiliations:** 1Division of Gastroenterology, Center for Autoimmune Liver Diseases, Department of Medicine and Surgery, University of Milano-Bicocca, 20900 Monza, Italy; alessio.gerussi@unimib.it (A.G.); c.caime@campus.unimib.it (C.C.); eleonora.binatti@unimib.it (E.B.); l.cristoferi@campus.unimib.it (L.C.); 2European Reference Network on Hepatological Diseases (ERN RARE-LIVER), San Gerardo Hospital, 20900 Monza, Italy; 3Department of Biomedical Sciences, Humanitas University, Via Rita Levi Montalcini 4, 20072 Pieve Emanuele, Italy; elvezia_maria.paraboschi@hunimed.eu (E.M.P.); claudio.cappadona@humanitasresearch.it (C.C.); rosanna.asselta@hunimed.eu (R.A.); 4Humanitas Clinical and Research Center, IRCCS, Via Manzoni 56, 20089 Rozzano, Italy

**Keywords:** chromosome X, non-coding RNA, autoimmunity, sex bias, somatic mosaicism

## Abstract

Primary Biliary Cholangitis (PBC) is a rare autoimmune disease of the liver, affecting mostly females. There is evidence that epigenetic changes have a pathogenic role in PBC. Epigenetic modifications are related to methylation of CpG DNA islands, post-translational modifications of histone proteins, and non-coding RNAs. In PBC, there are data showing a dysregulation of all these levels, especially in immune cells. In addition, epigenetics seems to be involved in complex phenomena such as X monosomy or abnormalities in the process of X chromosome inactivation, which have been reported in PBC and appear to influence its sex imbalance and pathogenesis. We review here historical data on epigenetic modifications in PBC, present new data, and discuss possible links among X-chromosome abnormalities at a genetic and epigenetic level, PBC pathogenesis, and PBC sex imbalance.

## 1. Introduction

Primary Biliary Cholangitis (PBC), previously known as Primary Biliary Cirrhosis, is a rare disease of the small intrahepatic bile ducts with a striking female predominance [1].

According to Orphanet, which performs a systematic review of registries, health institutes agencies, and published literature, PBC incidence is 3 per 100,000 and prevalence is 21.05 per 100,000 worldwide. Similar figures have been described for Europe (incidence: 2.57 per 100,000; prevalence: 25 per 100,000) [2].

The disease affects mainly women (90% of patients), and it frequently develops around 40–60 years of age. Despite the female predominance, men who develop PBC tend to have a more severe form of the disease [3,4]. The principal characteristics of PBC are cholestasis, anti-mitochondrial (AMA), or PBC-specific antinuclear (ANA) antibodies positivity and histological evidence of chronic, granulomatous, lymphocytic small bile duct cholangitis [5]. However, the clinical presentation of PBC can be variable, with many patients being asymptomatic and presenting only abnormal liver function tests at the time of diagnosis [6]. Among common symptoms, patients report fatigue, pruritus, hyperpigmentation, jaundice, xanthomas, xanthelasmas, xerosis, and dermatographism [6].

The disease is caused by an autoimmune reaction directed against the E2 component of the pyruvate dehydrogenase complex (PDC-E2), an oxidative phosphorylation enzyme [7], eventually leading to a chronic injury to the biliary epithelial cells (BECs). Despite the ubiquitous expression of this autoantigen, a targeted biliary injury is observed. Possibly, damaged BECs, sensitized to apoptosis, expose PDC-E2 within an apoptotic bleb, thus triggering the autoimmune response that causes the focused biliary injury [8]. The immunogenic complex is recognized by circulating AMA autoantibodies, which localize to the apical surface of BECs and associate with apoptosis [8]. The immune infiltrate in the portal tracts of patients is predominantly composed of CD4+ T cells, with fewer increases in CD8+ T cells, consistently with involvement of the adaptive immune system. CD4+ and CD8+ T cells directed against mitochondrial autoantigens are not detected in controls or patients with other liver disorders [9]. Other T cell subpopulations have also been implicated in the disease, including pro-inflammatory TH17 cells [8,10], Treg cells, promoting self-tolerance, and T follicular helper cells that facilitate antibody production [11].

Treatment is mainly aimed at counteracting cholestasis and associated inflammation, and at preventing the fibrosis progression. The FDA-approved medications used for the treatment of PBC are ursodeoxycholic acid and obeticholic acid [6]. To date, alternative strategies targeting immune injury with biologically based therapies have been unsuccessful [5].

From a genetic perspective, PBC is a complex trait and several genetic variants contribute to its genetic architecture [12]. In addition, several environmental factors have been called in action as triggers of the autoimmune cascade, including infectious agents and xenobiotics [13].

Epigenetics stands at the crossroad between the genome and the environment and there is increasing evidence about its key role in the pathogenesis of the disease [14]. In addition, there are new data regarding the role of the X chromosome (ChrX), both at the genomic and epigenetic level, that need attention.

This review deals with the available evidence on epigenetic modifications present in PBC, with a detailed focus on the role of such modifications related to chromosome X. Figure 1 summarizes known epigenetic abnormalities identified in PBC.

## 2. Traditional Pillars of Epigenetics

Epigenetics includes all modification of DNA or related factors that do not involve the DNA sequence and can be inherited [15]. Cellular heterogeneity and identity is controlled by epigenetic information since the genomic sequence is identical in all cells of the body [15,16]. There are three different types of epigenetic information: DNA methylation, post-translation changes of histones (e.g., acetylation, methylation, ubiquitylation, as well as histone variants), and non-coding RNAs (including microRNAs, miRNAs, and long non-coding RNAs, lncRNAs). The fourth pillar of epigenetics is chromatin organization (e.g., DNA accessibility and domain organization), yet in PBC, this topic has been barely explored; therefore, this review will focus only on the three other pillars.

### 2.1. DNA Methylation

DNA methylation, i.e., the addition of a methyl group, refers to a covalent change preferentially involving the nucleotide cytosine in CpG sites, which is inherited during cell division and typically generates gene silencing [15].

A study investigated the methylation pattern of the *cluster of differentiation 40 ligand (CD40L)* promoter, based on its crucial role in T-cell priming, T-cell activation, and immunoglobulin class-switch recombination [17]. After isolating CD4^+^ T cells from peripheral blood mononuclear cells (PBMC) and examining the promoter status via bisulfite sequencing, the authors identified a significantly reduction in its methylation level as compared to healthy controls [18]. This finding was specific for CD4^+^ cells, since no differences were found for CD8^+^ lymphocytes between PBC cases and controls. Consistently, levels of *CD40L* mRNA expression were higher in CD4^+^ T cells of PBC patients [18]. In PBC, macrophages kill cholangiocytes via a CD40-mediated mechanism [19], which points to the growing body of knowledge about the role of innate immunity in PBC [20]. 

In addition, Immunoglobulin M serum levels were negatively correlated with the promoter methylation patterns [18]. Immunoglobulin M (IgM) levels are often abnormally high in patients with PBC and represent a useful ancillary finding in the diagnostic pathway [21]; before the study by Lleo et al., little was known about the reason why PBC patients frequently show this serum change. There is no evidence supporting that PBC patients have different variants in the *CD40L* gene locus, which is located on ChrX [18,22]. Whether high IgM levels represent only a disease epiphenomenon is still to be ascertained.

A more recent approach adopted a genome-wide scan of methylation patterns in a privileged cohort of monozygotic (MZ) twins discordant for PBC. The working hypothesis was that the onset of PBC in only one twin may depend on epigenetic changes occurring over time. Regions with different methylation patterns were found especially on ChrX (51/60 differentially methylated regions), with hypermethylation rather than hypomethylation the common finding in PBC probands [23]. While the small sample size (3 MZ twins) prevents firm conclusions and might suggest that observed differences could be due to randomness, it is still interesting to notice that 14 out of 60 genes were also differentially expressed between PBC cases and controls and involved pathways potentially linked with PBC pathogenesis, such as IFN-associated proteins or Interleukin 6 [23]. The hypomethylation in the promoter of *CD40L* was not confirmed in this study; however, Selmi and colleagues analyzed PBMCs and not CD4^+^ and CD8^+^ cells, which might explain the different experimental finding. 

A similar study encompassing a chrX-wide scan of epigenetic marks on PBMC was performed in PBC cases and controls and identified a hypomethylation of the promoter of *C-X-C motif chemokine receptor 3* (*CXCR3)* [24]. Before this study, it was already known that *CXCR3*-positive cells were more often present in peripheral blood and in the liver of patients with PBC than in normal subjects [25], but no data on the epigenetic dysregulation were available. CXCR3 is a key chemokine in leukocyte trafficking, expressed mostly in activated T lymphocytes and Natural Killer (NK) cells [26]. On a similar note to what was found for *CD40L*, there is an inverse correlation between *CXCR3* expression in CD4^+^ cells and methylation levels [24]. The lack of PBC patients of male sex hampers to draw conclusions on the potential chrX-dependent sex differences in epigenetic abnormalities.

### 2.2. Post Translational Modifications to Histone Proteins

The DNA double helix is wrapped around proteins called histones. Several post-translational modifications affecting histones, including acetylation, methylation, phosphorylation, ubiquitylation, and sumoylation, act by modifying chromatin structure, recruiting remodeling enzymes that reposition nucleosomes. In this way, histone modifications can influence transcription and activate or silence genes by changing the DNA accessibility to transcription factors or enhancers [15].

T lymphocytes from patients with PBC have higher expression levels of *β-Arrestin 1 (βarr1)* than controls (either healthy ones or with chronic hepatitis B). In vitro studies of autoreactive T cells showed that *βarr1*, which is involved in T cell activation and has a pathogenic role in multiple sclerosis and other autoimmune conditions [27], is an epigenetic modulator. Chromatin immunoprecipitation (ChIP) assays showed that acetylation of histone H4 in the promoter regions of *CD40L*, *LIGHT* (also known as *tumor necrosis factor superfamily member 14*), *Interleukin 17 (IL17)*, *Interferon γ (IFNγ)*, *TNF-related apoptosis-inducing ligand (TRAIL)*, *Apo-2*, and *Histone deacetylase 7A (HDAC7A)* is under a dose-dependent control of *βarr1* in autoreactive T cells [27]. While for *CD40L*, *LIGHT*, *IL17*, and *IFNγ* the relationship is directly proportional (the higher the levels of *βarr1* the higher the acetylation), for *TRAIL*, *Apo-2*, and *HDAC7A*, the relationship is inversely proportional [27]. Importantly, *LIGHT* is critical in autoimmunity, since it is a co-stimulatory molecule vital for the control of T cell proliferation. *TRAIL* is vital for apoptosis induction, but it is involved in autoimmunity prevention through cell cycle arrest. *TRAIL* could have dual role in PBC: promoting apoptosis and the development of autoantibodies, but also the susceptibility to autoimmunity [23]. Finally, there is evidence that in B cells, histone modifications and differentially methylated CpG sites are mostly found in enhancers and promoters [28].

### 2.3. Non-Coding RNA Expression

Non-coding RNAs (ncRNAs) are RNAs that are not translated into proteins. Two main classes can be distinguished: small non-coding RNAs (when their length <200 nt; this class includes miRNAs) and lncRNAs (characterized by a length >200 nt). NcRNAs can regulate gene expression by different mechanisms: miRNAs, the most studied, regulate gene expression by translational repression or mRNA destabilization [29]. LncRNAs are, instead, a very heterogeneous class. By interacting with DNA, other RNAs, and proteins, they can modulate chromatin structure and the activity and the transcription of other genes, and affect each aspect of RNA post-transcriptional processing, including mRNA stability, splicing, transport, and degradation [30].

Several studies have shown dysregulation of specific miRNAs in PBC (Table 1). Despite this long list of observations, there has been only one study that incorporated one miRNA within a model of PBC pathogenesis. The classical view of PBC etiopathogenesis involves an environmental exposure (most likely a drug or an infectious agent) occurring in a subject with an unfavorable set of predisposing gene variants [12]. The autoimmune attack to biliary cells in PBC, following the breakdown of PDC-E2 immune tolerance, is considered an event occurring earlier than cholangiocytes abnormalities in the pathogenic cascade. Biliary epithelial cells in apoptosis release apoptotic bodies, which are only partially cleared by phagocytes in PBC, representing a potential powerful trigger for autoimmunity [20]. 

An alternative hypothesis suggests it would be more accurate to reverse the cascade of events based on the available experimental evidence [31]; defects in cholangiocyte bicarbonate secretion might have a more prominent role in the initiation phase. Cl^−^/HCO_3^−^_ anion exchanger 2 (AE2) is the chief protein involved in the homeostasis of the biliary epithelium, and it is located on the apical surface of lining cholangiocytes. Ae2-deficient mice resemble PBC in several features [32] and *AE2* gene expression is reduced in the liver and PBMC of patients with PBC [33]. *AE2 mRNA* is targeted by *miR-506*, which is upregulated in PBC but not in Primary Sclerosing Cholangitis, another autoimmune disease of the liver, or in healthy controls [34]. *Mir-506* binds the 3′ untranslated region of the *AE2* mRNA and prevents protein translation, while PBC cholangiocytes treated with anti-miR-506 have improved AE2 activity [34]. *mir-506* is also capable to alter mitochondrial energetic metabolism and promote apoptosis. In addition, pro-inflammatory cytokines (IL-8, IL-12, IL17, IL-18, and TNF-α), typically elevated in the liver of patients with PBC, are capable to stimulate the activity of the *miR-506* promoter, which in turn may activate immune cells; mir-506-overexpressing cholangiocytes do show increased expression of pro-inflammatory and pro-fibrotic markers [35]. Based on this evidence, Banales and colleagues suggested that breakdown of tolerance in PBC follows rather than precedes cholangiocyte secretory failure. Consistently with this theory, no genetic variants from genome-wide association studies in PBC have tagged the genetic region of the *AE2* locus [12], as the abnormality would be at the epigenetic level. Intriguingly, *miR-506* is located on the X-chromosome.

Concerning lncRNAs, few of them have been implicated in PBC. Zhang and colleagues identified *H19*, already known to be a regulator of liver development, as a key player in bile duct ligation-induced cholestatic liver injury [36]. Several studies have reported upregulation of this lncRNA in different hepatic cells, cholestatic mouse models, and human patients with PBC, as well as other cholestatic disorders as primary sclerosing cholangitis and biliary atresia. *H19* has multiple functions: in fact, it participates in different signaling pathways involved in the activation of macrophages, cholangiocytes, and hepatic stellate cell, and also functions as miRNAs sponge [37].

In addition, lncRNA *AK053349* was found increased in PBMCs of PBC patients, and it positively correlated with the Mayo risk score, a mathematical model predicting survival in non-transplanted patients, suggesting its possible relevance in the disease pathogenesis [31]. Interestingly, this lncRNA is one of the most strongly upregulated ncRNAs during CD8+ effector cell activation, and it partially overlaps a well-characterized cluster of miRNAs, whose forced overexpression in mouse lymphocytes results in autoimmunity, probably due to the activation of lymphocyte proliferation and survival programs after activation [38].

## 3. ChrX Monosomy and Inactivation

Inactivation of ChrX is the physiological phenomenon by which one of two ChrXs is silenced; the main goal is dosage compensation, i.e., to equalize ChrX gene expression between mammalian males and females [45]. This process is controlled by a lncRNA called *XIST* (*X-inactive-specific transcript*) RNA, which works together with some chromatin-modifying factors inducing heterochromatinization of ChrX [46].

Theoretically, if the 2nd ChrX was fully suppressed, X monosomy would not have negative consequences; this is the Lyon’s theory, that postulated that the 2nd ChrX is totally inert [47]. Yet, it is common evidence that germinal X monosomy (e.g., X0 in Turner syndrome) has negative effects on normal development, fertility and overall health of X0 individuals [48]. 

Importantly, under normal conditions, several genes located on ChrX escape silencing [49,50], leading to higher expression of X genes in females than males. In case of germinal X monosomy there could be haploinsufficiency for these genes that escape silencing, with potential negative effects on human health. A potential explanation for the developmental abnormalities in Turner syndrome is related to haploinsufficiency for pseudo-autosomal regions of sex chromosomes. These regions are located at the terminals of short (PAR1) and long (PAR2) arms of both X and Y chromosomes; due to technical reasons, the full sequencing of these sections of the sex chromosomes has been completed only recently [51]. PAR1 genes encode proteins involved in several functions such as transcriptional regulation, RNA splicing, signal transduction and others [49]. PAR2 seems to be specific of the human species and encode for few genes; of interest for immunity, *IL9R* is located on this region and seems to escape X inactivation [49]. Another region of interest involve all those genes that have X and Y homologs outside PAR1 and PAR2; some authors call them “pseudo-pseudoautosomal” genes [49]. Among others, *DEAD-Box Helicase 3HDAC7A (DDX3)*, *Eukaryotic Translation Initiation Factor 1A (EIF1AX)*, *jumonji*, *AT rich interactive domain (JARID)*, *Ubiquitously Transcribed Tetratricopeptide Repeat (UTX)*, *Ubiquitin Specific Peptidase 9 X-Linked (USP9X)* and *Zinc Finger Protein X-Linked (ZFX)* may have a significant role [49]. 

Another aspect that should be considered is the parental origin of the silenced ChrX. In males, ChrX is always derived from the mother, while females are mosaics. It turns out that in females 50% of cells express an active ChrX of mother origin and the other 50% express an active ChrX of paternal origin. There is evidence that DNA methylation of X genes can vary according to the parental origin; a higher expression of the X gene *toll-like receptor 7* (*Tlr7*) in XY mice with experimental autoimmune encephalomyelitis (a model system of multiple sclerosis) than XX littermates can only be explained by epigenetic modifications in the parental germ line (“parental imprinting”) [52]. Indeed, dosage effect would justify higher expression in XX and not XY individuals. This finding was validated in subsequent studies, where the potential confounding due to sex hormones advocated the use of a model with Y^−^ chromosome, i.e., a Y chromosome with the deletion of the gene responsible for testicular development. It is important to stress that investigators found also an imbalance due to escape genes (some genes were more expressed in the XX than in the XY^−^ mice) together with the mentioned higher expression in XY^−^ for *Tlr7*, underlining again the contribution from several mechanisms to sex bias [53]. Yet, the epigenetic difference is still speculative and indirect; unfortunately, the direct study of differences in methylation patterns is not trivial because the DNA of the inactive X is highly methylated due to X inactivation [50]. 

In the same study from Golden and colleagues, they overcame this hurdle by generating a mouse model with only one ChrX (X-monosomic mice): in this way no X inactivation is present and ChrX derives either from the mother or the father [53]. Investigators showed that parental imprinting in DNA methylation of ChrX generate differences between males and females in CD4^+^ T lymphocytes stimulated by autoantigens [53]. Targeted bisulfite sequencing of the region upstream of the *Tlr7* transcriptional start site showed that CpG sites were more methylated on the paternal X, consistent with higher *Tlr7* RNA expression in the monosomic mice with the maternal X. The difference in methylation was not restricted to *Tlr7* but involved several other X genes and were not strain-specific. Authors conclude that sex bias in autoimmunity depends on parental imprinting on genes not escaping inactivation, and dosage effect from escape genes.

Other important mechanisms could be related to changes in the physical properties (size, shape, and circularity) of ChrX in monosomic cells [48,54] and abnormalities of the epigenetic regulation of gene expression on the active ChrX in X-monosomic cells [55]. The fraction of the active ChrX is larger in monosomy X karyotypes by 12% [54], while no differences in circularity, a measure of chromatin compaction, calculated as a ratio between area and perimeter, have been shown in ChrX from primary cells of individuals with germinal X monosomy [53]. 

Together with ChrX inactivation, there is evidence that evolution has generated a mechanism to balance the risk of haploinsufficiency in male mammals. This goal is achieved by doubling the global expression levels of the active ChrX, especially in brain tissues; while in *D. melanogaster* the upregulation of the X occurs only in males, in human and mouse tissues the dosage compensation is present in both sexes [54]. In fact, by comparing the global transcriptional output from the X chromosome with that of autosomes using microarray data, it was possible to calculate the mean fluorescence intensity of X-linked versus autosomal genes for each set of arrays hybridized with labeled cDNA in different tissues [54]. From this analysis it was evident that the global transcriptional output from the X chromosome is doubled in both species to achieve dosage compensation. A similar result have been obtained in tissues from other organisms [56].

Evidence supports that dosage compensation operates at the transcriptional level, but also post-transcriptional and epigenetic changes might be involved [54]. It turns out that abnormalities in the control of the gene expression of the active ChrX may also partially justify the negative health effects of X monosomy. 

Apparently, in females there is a combination of doubling the transcriptional output from the active X and X inactivation. Since at least 23% of the genes on ChrX escape silencing based on more recent data [50], one could wonder how it is possible to maintain a balance. Earlier experimental evidence seemed to suggest that only a tiny minority of escape genes have a significant increase in expression in female subjects [55]. Yet, more recent data do show that sex-biased expression of at least 60 genes is dependent on ChrX inactivation contributing to phenotypic divergence and variability [50]. Importantly, it is worth to underline that sex-related phenotypic variance is not restricted to ChrX. Sex-biased expression of autosomal genes is also quite relevant and derives mostly from hormone-related transcription factor regulation and sex-differentiated distribution of epigenetic marks [57].

X monosomy in somatic cells has been associated with ageing and multiple diseases, including autoimmune conditions [58,59]. Monosomy for ChrX in somatic cells is more pronounced in lymphocytes as compared to other cells, and it is dependent on telomere length (chromosomes with shorter telomeres have higher loss rates [60]. Our group contributed with seminal experimental works describing this phenomenon in PBMC of female patients with PBC [61,62]. After adjustment for age, women with PBC do show increased rate of X monosomy in PBMC [61]; of note, ChrX loss involves always the inactive homologue, suggesting that X monosomy does not follow a non-random pattern of ChrX inactivation [62]. Interestingly, many micronuclear buds (see below) were visible all around the main nucleus of PBMC from PBC patients; inside these atypical structures, one or more X centromeric region *HOR sequence on the X chromosome* (*DXZ1*) positive signals were identified, revealing that the exclusion of whole or damaged ChrX from the main nucleus into micronuclei occurs more frequently in PBC samples than in controls [61]. 

The formation of micronuclei has been associated with complete or partial monosomy [63]. Micronuclei are abnormal extranuclear small bodies, budding from the main cell nucleus and located nearby it. These structures are characteristic of cells with some sort of DNA damage [64]. Physical isolation of chromosomes in the micronuclei might explain the confined DNA lesions observed in chromothripsis [65]. In chromothripsis, thousands of chromosomal rearrangements and variations in DNA copy number happen at the same time in a constrained genomic region of one or few chromosomes. This phenomenon is known to be involved in cancer onset and congenital diseases [66]. The mechanism behind chromothripsis is unknown, but it might occur through the partitioning of a chromosome in micronuclei as a consequence of a defective mitosis [67]. Inside micronuclei, DNA replication is slower than in the main nucleus, and poorly replicated chromosomes become crushed and broken. If chromothripsis occurs, chromosome segments are aberrantly joined together to give rise to rearranged products that can subsequently be reincorporated into the main nucleus of a daughter cell [67].

Under physiological conditions, ageing is associated with an increase in both X-chromosome loss and micronuclei formation, mainly observed in peripheral lymphocytes [68,69]. Despite historically associated with ageing and cancer, micronuclei are now considered an important player in autoimmunity. Theoretically, micronuclei could be responsible for the initiation of the autoinflammatory cascade: chromothripsis causes DNA leakage from micronuclei and activation of the Cyclic GMP-AMP Synthase (cGAS)-Stimulator Of Interferon Response CGAMP Interactor 1 (STING) axis that triggers innate immunity [58]. A higher threshold of activation of cellular DNA sensors in T lymphocytes, due to genetic predisposition to autoimmunity [12], could result in stronger autoimmune-responses. For a thorough and detailed description of mechanisms linking micronuclei formation and inflammation see [58]. 

To sum up, X monosomy could increase the risk of autoimmune phenomena because the entrapment of the inactive homologue within micronuclei could cause functional haploinsufficiency of regulatory genes but also because micronuclei themselves could ignite inflammatory reactions [58].

## 4. XWAS Results and How They Associate with Epigenetics

Several genome-wide association studies (GWAS) have been performed to elucidate the genetic architecture of common variants in PBC and identified more than 50 loci contributing to the disease risk [12]. In general, GWAS analyses often focus on autosomes, and polymorphisms mapping on ChrX are usually not included or discarded from the analysis to avoid to account for the analytical problems due to its unique inheritance patterns between males and females [70]. Therefore, the role played by ChrX in many complex diseases and traits remains largely unknown. Considering that ChrX can be regarded as an immunologic chromosome, since it contains the largest number of immune-related genes compared to other chromosomes [71], its analysis in the context of PBC is fundamental to uncover genes possibly contributing to the disease. Indeed, a thorough assessment of the contribution of ChrX has only been investigated recently. Our group has identified a new locus, tagged by the rs7059064 polymorphism, associated with the disease and localized on ChrX [22]. Interestingly, this polymorphism underlies a unique linkage disequilibrium block including 7 genes (*Translocase Of Inner Mitochondrial Membrane 17B (TIMM17B)*, *Polyglutamine Binding Protein 1 (PQBP1)*, *Pim-2 Proto-Oncogene*, *Serine/Threonine Kinase (PIM2)*, *Solute Carrier Family 35 Member A2 (SLC35A2)*, *OTU Deubiquitinase 5 (OTUD5)*, *Potassium Voltage-Gated Channel Subfamily D Member 1 (KCND1)*, and *GRIP1 Associated Protein 1 (GRIPAP1)*, as well as a superenhancer targeting all these genes and *FOXP3*, the main T-regulatory cell lineage specification factor [22]. Superenhancers are clusters of enhancers that act as a switch to determine cell fate and identity, and they are characterized by a specific epigenetic signature: nucleosomes binding to enhancer flanking regions are enriched in histone H3 lysine 4 monomethylation (H3K4me1) and histone H3 lysine 27 acetylation (H3K27ac) [72]. More recently, it was demonstrated that enhancer regions support transcription and give rise to non-coding enhancer RNAs (eRNAs), that have become a hallmark of active enhancers. The enhancer identified by the X-chromosome analysis is actively transcribed and significantly expressed in thymus, spleen, and white blood cells. Moreover, it is enriched in binding sites for immune-related nuclear factors, thus suggesting a possible involvement in an immune-mediated regulation of target genes. Several studies have investigated the role of superenhancers especially in autoimmune diseases [73]. In particular, an enrichment of disease-associated SNPs was observed in superenhancer domains of disease-relevant cells and tissues, thus suggesting that a modified expression of cell identity genes may contribute to disease pathogenesis [74]. This enrichment was demonstrated, among autoimmune disorders, also for PBC, thus highlighting the role of epigenetics in the predisposition to the disease.

To better analyze the ChrX epigenetics signature in PBC, we further explored the association signals resulting from the work by Asselta and collaborators [22] especially focusing on methylation and ncRNAs. We selected the genes pinpointed by those SNPs that displayed a nominal *p* < 0.0005 in association analyses (for details see [22]), and then compared them with a manually curated list of differentially methylated genes in PBC derived from the literature [24,31,75]; a bootstrapping approach allowed us to demonstrate that there is a significant enrichment (*p* = 0.013) in differentially methylated genes among those showing a suggestive signal of association with PBC (Table 2).

Moreover, we also searched for possible enrichments in terms of non-coding elements, i.e., lncRNAs, miRNAs, and circular RNAs (circRNAs) in the X-chromosome PBC-associated regions (see Appendix A for details). This analysis revealed the presence of 22 lncRNAs, 362 circRNAs, and 9 miRNAs partially or totally overlapping the analyzed genomic regions (Table 3). To test for possible significant enrichments in these non-coding elements, the same workflow was applied to randomly selected chrX regions (random sets). To this aim, each random set was composed of the same number of regions, and the same genomic width as the PBC-related set. The process was repeated 1,000 times. The pipeline identified, on average, 18.6 lncRNAs, 177 circRNAs, and 10 miRNAs (Table 3). Comparing the random set results with the PBC dataset, we obtained the same or a larger number of lncRNAs, circRNAs, and miRNAs in the 24.5%, 5%, and 46.5% of the iterations, respectively, thus indicating a significant enrichment in circRNAs (*p* = 0.05) in the PBC dataset. In this framework, it’s interesting to note that circRNAs expression was found dysregulated in plasma of PBC patients [76]. The authors showed that a particular circRNA, *hsa_circ_402458*, might target a miRNA, *miR-522-3p*, potentially implicated in regulating chronic inflammatory disorder [76].

## 5. The Role of the Environment

Environmental factors seem to account for >80% of the risk of developing human disease [15]. Among others, there is evidence supporting the role of infectious agents, xenobiotics, and dietary habits in shaping the risk of diseases; in addition, microbiome variation is another important area of research. PBC makes no exception and there are data for some of these potential factors [14]. 

Epidemiological studies have consistently identified a link between PBC and urinary tract infections promoted by *Escherichia coli* [13,77,78]. Remarkably, the human PDC-E2, which is the immunodominant epitopes of anti-mitochondrial antibodies (AMAs), overlaps from a molecular point of view the *E. coli* PDC-E2 [79]. Molecular mimicry has been called in action also for another bacterium, Novosphingobium aromaticivorans, with data supporting a much higher reactivity to PBC patient sera than *E. coli* [80]. There is also a pathogenic theory proposed by Mason and colleagues that PBC is driven by a human betaretrovirus, since nucleic adic sequences have been cloned from the lymph nodes of PBC individuals [81].

There is large amount of data that support the role of xenobiotics as triggers of PBC. In case-control studies evaluating different factors related to lifestyle of subjects with PBC the use of nail polish was associated with higher risk of PBC [78]. Landmark studies evaluating the geographical association between the place where patients lived and factories has revealed that individuals with PBC concentrated close to toxic waste sites [82,83,84]. The most recent study evaluating spatial clustering of PBC and primary sclerosing cholangitis has shown that PBC patients cluster in urban areas with previous history of coal mining and high levels of cadmium, as compared to patients with PSC more aggregated in rural areas [85]. Molecular mimicry with xenobiotics has also been investigated in detail, revealing that 2-octynamide and 2-nonyamide have high reactivity with PBC sera. Cosmetics, lipsticks and chewing gums all contain derivatives of 2-octynoic acid. Importantly, the conformation of PDC-E2 inner lipoyl domain is altered when conjugated with 2-ocynoic acid, supporting the notion that xenobiotics alter lipoamide which becomes the initial target of autoimmunity [86]. More recent evidence has identified a ionic liquid present in soil extract from around an urban landfill as structurally similar to lipoic acid; mitochondrial effects have been revealed in vitro on hepatic progenitor cells and human liver cholangiocytes [87].

Dysbiosis has been studied and characterized in PBC [88]. If compared to healthy subjects, PBC patients have lower bacterial diversity, with reduced abundance of Clostridiales and increased abundance of Lactobacillales [88]. Patients with poor response to first-line therapy (Ursodeoxycholic acid) have lower abudance of the genus Faecalibacterium [88]. Whether dysbiosis is causative or a consequence of cholestasis is still to be determined; in fact, progression of the disease may influence per se the composition of gut microbiota and the relationship between gut microbiota and bile acids is not unidirectional [89].

## 6. Future Perspectives and Final Remarks

Over the last decade, a significant number of genome-wide epigenetic studies have been performed in PBC, targeting different cell types, with conflicting results and heterogenous design. Yet, these studies had the merit to spot a light on the role of epigenetics in this rare condition, which has a strong genetic background but also several alleged environmental triggers that may ignite its pathogenic cascade. In parallel, the body of evidence supporting the role of chrX in PBC has continued to grow, moving from early signals of somatic mosaicism to the more recent and robust data coming from a large meta-analysis focusing on chrX. The unique features of chrX have represented a hurdle that has discouraged its dissection in autoimmunity for too long. Our group has shown that its study can bring important insights on abnormalities at the genetic and epigenetic level, which prompt further functional validation in different preclinical models. In addition, the identification of *miR-506* and the finding of its upregulation in PBC has paved the way for a re-thinking of the established theories of PBC etiopathogenesis. Moreover, this line of experimental evidence has clearly revealed the importance of studying disease-related tissues on top of blood.

While speculative, a challenging but attractive future perspective might be the development of epigenetic modifiers. 

To conclude, the study of epigenetic abnormalities in PBC may well inform disease pathogenesis, represent a way to identify novel biomarkers to assist risk stratification, and, potentially, help to develop new targeted drugs.

## Figures and Tables

**Figure 1 ijms-23-04873-f001:**
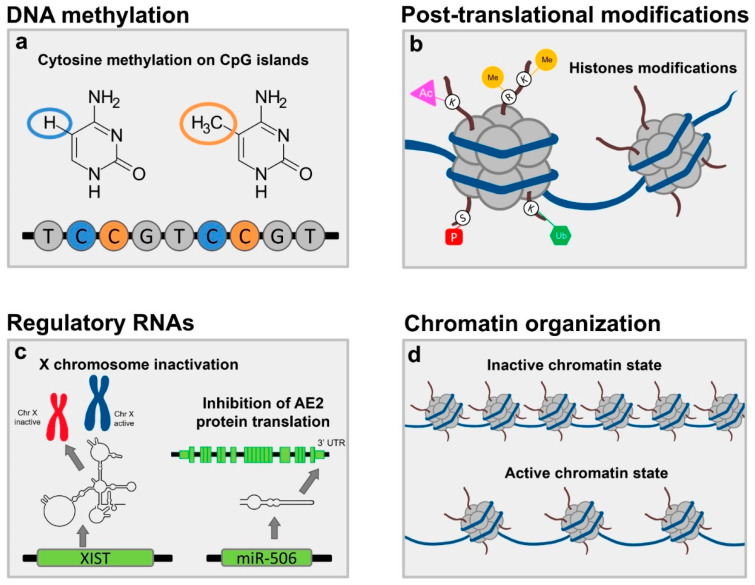
Epigenetic modifications in PBC. (**a**) DNA methylation refers to the transfer of a methyl group onto the C5 position of the cytosine to form 5-methylcytosine in CpG sites, and typically, it induces gene silencing. In PBC cases, a significantly reduction in methylation level of *CD40L* and *CXCR3* promoters was observed in CD4+ cells, resulting in a higher expression of these genes. (**b**) Histone modifications, e.g., acetylation, are post-translational modifications that affect histones and impact their interaction with DNA. PBC patients have high expression levels of βarr1 in T lymphocytes, which controls acetylation of histone H4 in the promoter regions of several genes (*CD40L*, *LIGHT*, *IL17*, *IFN-γ*, *TRAIL*, *HDAC7A*). (**c**) Non-coding RNAs are RNAs that do not encode proteins. In PBC, overexpression of microRNA *miR-506* results in the dysregulation of *AE2*, pro-inflammatory, and pro-fibrotic genes expressions. Additionally, *XIST RNA*, a long non-coding RNA, controls the inactivation of one of two ChrXs in females for dosage compensation. Some genes with immune-related functions and located on the inactive ChrX (Xi) are known to escape XCI (e.g., *CD40L*, *CXCR3*, *TLR7*). This suggests a possible link between the altered expression of genes from the Xi and a greater susceptibility to autoimmunity disorders. (**d**) The center of epigenetic gene regulation is represented by the chromatin organization. Chromatin is a complex of DNA and proteins, and it is organized in euchromatin and heterochromatin. These two different structures influence the accessibility of promoter regions to transcription factors. There is a lack of data in PBC.

**Table 1 ijms-23-04873-t001:** Dysregulation of miRNAs in Primary Biliary Cholangitis.

miRNAs	Tissue	Notes	Reference
miR-122a, miR-26a, miR-328, miR-299-5p	Liver	miR-122a/miR-26 DOWN miR-328/miR-299-5p UP	Padgett et al., 2009 [39]
miR-506	Liver	miR-506 UP	Banales et al., 2012 [34]
miR-15a-5p, miR-20a-5p, miR-140-3p, miR-106b-5p, miR-3654, miR-181a-5p	PBMCs	abnormal expression of 17 miRNAs that control cell differentiation and signal transduction	Qin et al., 2013 [40]
hsa-miR-505-5p, hsa-miR-141-3p, has-miR-26b-5p	Sera	hsa-miR-505-3p/miR-197-3p DOWN	Ninomiya et al., 2013 [41]
hsa-miR-122-5p, hsa-miR-141-3p, hsa-miR-26b-5p	Sera	disease biomarkers	Tan et al., 2014 [42]
miRNA-122, miRNA-378, miRNA-4311, miRNA-4714-3p	Sera	risk stratification	Sakamoto et al., 2016 [43]
miR-92a	Sera, PBMC	pathogenesis (Th17 cell differentiation)	Liang et al., 2016 [44]

**Table 2 ijms-23-04873-t002:** Characteristics of differentially methylated genes in PBC showing signals of association with the disease at *p* < 0.0005.

*Gene*	*Study Design*	*Analyzed Samples*	*Methylation Status*	*Expression in* *Liver/Blood* *(TPM)*
*PIN4*	MZ twins	serum	Variable	5.89/0.94
*NHS*	MZ twins	serum	Hyper-methylated	0.12/0.25
*IL1RAPL2*	cases vs. controls	serum (CD14+)	Hyper-methylated	0.17/0.10
*SHROOM2*	cases vs. controls	serum (CD4+)	Hypo-methylated	1.89/0.030
*ATP6AP2*	cases vs. controls	serum (CD4+)	Hypo-methylated	33.37/55.74
*PQBP1*	cases vs. controls	serum (CD4+)	Hyper-methylated	35.06/31.37
*MAGEB2*	cases vs. controls	serum (CD8+)	Hypo-methylated	Not expressed
*NSDHL*	cases vs. controls	serum (CD14+)	Hypo-methylated	20.97/5.21
*PIM2*	cases vs. controls	serum (CD14+)	Hyper-methylated	4.80/60.37

Data on methylation status were retrieved from the literature [24,31,70]; data on expression status in liver and whole blood were retrieved from the GTEx portal (https://gtexportal.org/home/, accessed on 1 December 2018).TPM corresponds to transcripts per million reads (RNAseq data) normalized for transcript/gene length.

**Table 3 ijms-23-04873-t003:** Enrichment in non-coding elements in PBC.

Set	Regions (n)	Regions § (kb)	DNA Content (Mb)	LncRNAs (n)	CircRNAs (n)	MiRNAs (n)	SEs (n)
PBC set	62	150.3	9.3	22	362	9	351
SD	-	116.9	-	-	-	-	-
Random sets *	62.0	152.8	9.3	18.6	177.0	10.0	370.1
SD	-	-	-	4.6	78.0	7.2	51.4
% **	-	-	-	24.5	5.1	46.5	63.6
*p* value	-	-	-	0.25	0.05	0.47	0.64

The analyses were performed on the PBC-associated loci (upper part) and on 1000 random sets (lower part). § Average length of the genomic regions comprising the SNPs of interest. * For the random sets analysis, the average values calculated on 1000 iterations are indicated. ** % of times in which the same or a larger number of lncRNAs, circRNAs, and miRNAs was obtained in the 1000 iterations as compared to the PBC dataset. *p* values were calculated as described [73]; significant *p* value is indicated in bold. CircRNA, circular RNA; lncRNA, long non-coding RNA; miRNA, microRNA; *n*, number; SD, standard deviation; SE, super enhancer.

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
