# Peer review of "The Role of Epigenetics in Primary Biliary Cholangitis"

_ijms, 2022, doi:10.3390/ijms23094873_

Round 1

Reviewer 1 Report

In the article entitled: The role of Epigenetics in Primary Biliary Cholangitis authors have taken the effort to connect the epigenetic data with the classical medical point. The hepatic diseases are one of the main problems of early diagnosis and medical treatment. Bile duct and gallbladder malfunction are a frequent female problems, due to the feeding scheme, genetic inheritance, contraception treatment, etc. The author in their manuscript put their attention to epigenetics. They properly describe several epigenetic points: the DNA methylation, post-transcriptional modifications to histone proteins, ChrX gene expression, genome-wide association studies and from this point the article is valuable. However, I did not find the description of nutrition influence on the genome, strictly medical description and the manifestation of the discussed problem, the influence of PBC on patient quality of life. Moreover, the authors totally omit DNA damage repair machinery, level of DNA damage, etc. It is important to describe the above in the context of DNA methylation context. Additionally, I have found nothing about sparing genotype/phenotype which is important in my opinion when we discuss “liver problems”. In conclusion, the article is well written however is hard to read. It is difficult to follow the author's idea.

The article can be accepted for publication after the above remarks' consideration.

Reviewer 2 Report

In the review entitled “The role of Epigenetics in Primary Biliary Cholangitis”, the authors summarize the pathogenic role of epigenetic changes in Primary Biliary Cholangitis (PBC). In particular, they focused on X chromosome inactivation, which have been reported in PBC and appear to influence its sex imbalance and pathogenesis.

The review is well-organized, clear and simple to read and well focused on the concepts it aims to dissect.

I think that before it can be considered for publication, the review just needs:

  • To add an Abbreviations Section, in order to list the acronyms reported within the manuscript and then to report the complete name the first time it appears within the text (ie.CD40L at lane 106)

In conclusion, I think that the review can be considered for publication after a slight revision and the adding of some figures within the manuscript and not only in the supplementary material.
